# Graphene Oxide Nanofiltration Membranes Containing Silver Nanoparticles: Tuning Separation Efficiency via Nanoparticle Size

**DOI:** 10.3390/nano10030454

**Published:** 2020-03-03

**Authors:** Kun Yang, Lin-jun Huang, Yan-xin Wang, Ying-chen Du, Zhi-jie Zhang, Yao Wang, Matt J. Kipper, Laurence A. Belfiore, Jian-guo Tang

**Affiliations:** 1Institute of Hybrid Materials, National Center of International Research for Hybrid Materials Technology, National Base of International Science & Technology Cooperation, College of Materials Science and Engineering, Qingdao University, Qingdao 266000, China; karrenyang@126.com (K.Y.); yanxin_2008@126.com (Y.-x.W.); duyingchen@126.com (Y.-c.D.); 2018020367@qdu.edu.cn (Z.-j.Z.); wangyaoqdu@126.com (Y.W.); 2Department of Chemical and Biological Engineering, Colorado State University, Fort Collins 80521, UK; matthew.kipper@colostate.edu (M.J.K.); belfiore@engr.colostate.edu (L.A.B.)

**Keywords:** graphene nanofiltration membrane, siver nanoparticles, size control

## Abstract

Three types of graphene oxide/silver nanoparticles (GO/AgNPs) composite membranes were prepared to investigate size-effect of AgNPs on nanofiltration ability. The size of AgNPs was 8, 20, and 33 nm, which was characterized by UV-visible spectroscopy and transmission electron microscopy. The morphology and structure of GO and GO/AgNPs composite membranes were characterized by atomic force microscopy, scanning electron microscopy, and X-ray diffraction. The filtration performance of membranes were evaluated on a dead-end filtration device. When the size of AgNPs is 20 nm, the GO/AgNPs composite membrane has the highest water flux (106.1 L m^−2^ h^−1^ bar^−1^) and rejection of Rhodamine B (RhB) (97.73%) among three types of composite membranes. The effect of feed concentration of dye solution and the flux of common solvent was also investigated. The mechanism was discussed, which demonstrated that both interlaying spacing and defect size influence the filtration ability of membrane, which is instructive to future study.

## 1. Introduction

Industrial sources of water pollution include emissions of heavy metal ions, organic dyes, and other organic solvents [1,2,3]. Organic dyes have been widely used in many industries, including printing, paper, textiles and leather production [4,5]. Wastewater containing organic dyes must be treated before being released, due to the stability and toxicity of dyes [6]. Nanofiltration (NF), one type of pressure driven membrane technology, is a promising strategy for addressing this problem [7]. A suitable membrane material for nanofiltration must have excellent mechanical properties, good permeability, high selectivity for some organic dye molecules, and chemical stability [8]. Though conventional polymeric NF membranes have many advantages, such as good separation performance, easy-fabrication, and low-cost, they have limitations including poor chemical stability and membrane fouling [9,10,11].

Since its isolation and characterization in 2004, graphene has drawn great attention from researchers worldwide for its good mechanical properties [12], large surface area [13,14], and quantum Hall effect [15,16,17,18]. Graphene oxide (GO), one derivative of graphene, has recently been studied as a membrane material for its easy preparation and chemical stability [19,20]. The large-scale preparation of GO can be achieved by chemical exfoliation of graphite directly [21]. GO membranes present oxygen containing, negatively charged functional groups, enabling them to reject negatively charged organic dyes [22]. 

Joshi et al. reported that ions smaller in size than nanochannels can easily pass through GO membranes, and that size exclusion is the dominant sieving mechanism [23]. Tuning the interlayer spacing of GO sheets is a promising strategy for improving the water flux of GO membranes [24]. Interlayer spacing can be altered via the incorporation of silver nanoparticles (AgNPs), to make GO/AgNP composite membranes [25,26]. Sun et al. [27] investigated the influence of the thickness of GO/AgNPs membrane to permeability. The result showed that the water flux increased with reduction of the membrane thickness. They also investigated the water contact angle of GO and GO/AgNPs membrane. The result showed that the water contact angle of GO/AgNPs membrane was larger than that of GO membrane, which indicated GO/AgNPs membrane is more hydrophobic. Liu et al. investigated influence of membrane thickness to rejection to RhB. The result showed membrane thickness positively correlated to rejection [28]. These studies indicate membrane thickness is critical to nanofiltration performance. Chen et al. [29] found a negative correlation between the size of AgNPs in GO@PEG composite membranes and antibacterial activities. However, there are few studies about how the size of AgNPs on GO membranes effects the permeability and rejection of organic dye.

In our previous work, graphene-based hybrid materials, such as GO/rare-earth materials [14], graphene/silver hybrid membranes [28], GO/polyacrylamide composite membranes [7], and graphene-gold nanoparticles membranes [30], have been reported. We have investigated the influence of the amount of AgNPs on membrane nanofiltration performance [28]. The fabrication method of different-sized AgNPs is mature [29]. Therefore, in this work, we chose AgNPs as spacer, and successfully fabricated three types of GO/AgNP composite membranes. The size of AgNPs loaded on GO membrane was 8 nm, 20 nm, and 33 nm respectively. The water flux and rhodamine B (RhB) rejection of three types of GO/AgNP composite membranes and GO membrane and their relationship with membrane thickness was investigated. We also investigated the relationship of these features with the size of AgNPs and proposed the separation mechanism. This work is helpful for understanding the separation mechanism of GO/AgNPs composite membranes, and reveals significant research questions for future study. 

## 2. Experimental Section

### 2.1. Materials

Natural flake graphite (99%) with average particle sizes of 300 mesh was purchased from Qingdao Tianheda Graphite Co., Ltd (Qingdao, China). H_2_SO_4_ (98 wt%) and H_2_O_2_ (30 wt%) were purchased from Zhiyuan (Tianjin, China) Chemical Reagent Co., Ltd. Silver nitrate (AgNO_3_), sodium citrate dihydrate (C_6_H_5_Na_3_O_7_·2H_2_O), ethanol (C_2_H_5_OH), methanol (CH_3_OH) and ethylene glycol ((CH_2_OH)_2_) were purchased from Sinopharm Chemical Reagent Co., Ltd., Shanghai, China. Rhodamine B (RhB) and methylene blue (MB) were purchased from Macklin Reagent Co., Ltd.,Shanghai, China. Poly(vinyl pyrrolidone) (PVP, (C_6_H_9_NO)_n_) with average molecular weight 2.4 × 10^4^ g/mol was purchased from Shanghai Aladdin Bio-Chem Technology Co., Ltd, China.

### 2.2. Preparation of GO/AgNPs Composite Membrane

GO was prepared by the improved Hummers’ method from natural flake graphite mentioned above. 25 mg GO was dissolved in 50 mL ultra pure water followed by sonication in Ultrasonic cell grinder for 4 min. 25 mg AgNO_3_ was dissolved in 2 mL ultra pure water and then added into GO solution. 4.2 g PVP was dissolved in 20 mL ultra pure water. Then, 1 mL PVP solution was added into the reaction solution to prevent the aggregation of silver nanoparticles. After sonication for 20 min, the mixture was heated to boiling, and then 150 mg sodium citrate dissolved in 10 mL ultra pure water was added to the solution. After boiling under reflux for 40 min, protected from light, GO/AgNP composite was prepared, and the size of the AgNPs was 8 nm. The concentration of AgNO_3_ can influence the size of AgNPs. 50 mg and 100 mg AgNO_3_ were dissolved in ultra pure water and then added to the composite solution, respectively. After boiilng under reflux for another 40 min, similar samples were prepared using AgNPs of 20 nm and 33 nm (Scheme 1). The image of composite solution was shown in Appendix A GO/AgNP composite membranes was prepared by the vacuum filtration method using equipment shown as Appendix A. A 0.22-μm cellulose acetate was used as support the membrane, which will be part of the final product. The composite solution was filtrated through cellulose acetate membrane under a transmembrane pressure of 1 bar (0.1 MPa), and then dried the composite membrane in air. The image of composite membrane is shown in Appendix A.

### 2.3. Performance Evaluation of GO/AgNPs Composite Membrane

A home-made dead-end filtration device was used to measure the membrane performance. The water flux values, which can be used to measure permeability of the membrane, were calculated according to:(1)$$J=\frac{V}{A\times t\times P}$$
where *V* (L) is the volume of water, *A* (m^2^) is the effective filtration area of the membrane, *t* (h) is the filtration time, and *P* (bar) is the transmembrane pressure.

Rhodamine B (RhB) and methylene blue (MB) were used to characterize the rejection performance of composite membrane. The rejection can be calculated using the following equation:(2)$$R=\left(1-\frac{A_{2}}{A_{1}}\right)\times 100\%$$
where *A*_2_ is the absorption of the filtrate, and *A*_1_ is the absorption of the original solution. 

### 2.4. Characterization of GO/AgNPs Composite and Membrane 

The oxygen containing groups on GO and GO/AgNPs were characterized by Fourier-transform infrared spectroscopy (FTIR; MAGNA-IR 550, Thermo Nicolet Corporation, Beijing, China). The oxidized sp^2^ structure was characterized by raman spectroscopy (Thermo Scientific DXR 2xi, Shanghai, China). The morphology and structure of GO and GO/AgNP composite membranes were characterized by scanning electron microscopy (SEM; JEOL-6460, JSM-750, Beijing, China) and atomic by force microscopy (AFM; Bruker Bioscope Resolve, Beijing, China). The presence of AgNPs on GO was confirmed by UV-visible spectroscopy (UV-Vis; UV-1700, Shimadzu, Japan) within the wavelength range with 200–700 nm. The morphology of GO sheets and the size of AgNPs were evaluated by transmission electron microscopy (TEM; JEM-1200EX, Beijing, China). The lattice spacing of AgNP was determined by high-resolution transmission electron microscopy (HRTEM). The XRD pattern was obtained with X-ray diffraction (XRD) using a Bruker D8 Advance diffractometer (Bruker, Beijng, China). 

## 3. Result and Discussion

### Characterization of Composites and Membranes 

The images of GO and GO/AgNPs composite solution and membrane were shown in Appendix A, respectively. FTIR spectra of GO and GO/AgNPs composite is shown in Figure 1a. The peak at 3430 cm^−1^ corresponds to −OH stretching band. The peaks at 1740, 1610, 1230 and 1050 cm^−1^ are assigned to C=O stretching vibration, C=C vibration, C−O−C stretching and C−O stretching, which indicate successful synthesis of GO. Some prominent peaks shifted after AgNPs modification. Because of the strong interaction between Ag^+^ ions and carboxyl groups on the edges of GO sheets, the intensity of −OH decreases sharply, and both the intensity and peak position of C=O group changed, as shown in the FTIR spectra of GO/AgNPs composite. The intensity and peak position of C−O−C and C−O groups also changed, which indicates that the vast oxygen-containing functional groups contribute to the proper decoration of AgNPs on GO sheets. Compare to GO, GO/AgNPs composite have obvious change in peak position and intensity, which shows composite does not produce new chemical bonds, but due to the electrostatic adsorption. Figure 1b shows Raman spectra of GO and GO/AgNP composites. Comparing to GO, the D and G band obviously enhanced after AgNP modification, because of surface enhanced Raman scattering from the forceful local electromagnetic fields of AgNPs along with the plasmon resonance. The intensity ratio of GO/AgNPs composite (I_D_/I_G_ = 1.02) is also higher than that of GO (I_D_/I_G_ = 0.92), indicating a decrease in the average crystallite size of GO−AgNPs composite, due to more defects in graphene composite material after AgNPs modification. [27] The degree of disorder decreased because of the average crystallite size of GO-AgNPs composite decreasing. I_D_/I_G_ ratio is inversely proportional to the average crystallite size and the degree of disorder.

A TEM image of GO is shown in Figure 2a. The TEM image and size distribution of three types of GO/AgNP composites are shown in Figure 2b–d,f–h. The sizes of AgNPs loaded on the three types of GO/AgNP composites are 8 nm, 20 nm and 33 nm, respectively, as shown in Figure 2f–h, which is consistent with TEM image (Figure 2b–d). Therefore, we called them GO-8, GO-20 and GO-33. An HRTEM image of AgNPs loaded on a GO sheet is shown in Figure 2e. The inter-planar distance between the fringes is 0.24 nm, which is corresponds to the (200) lattice plane of Ag. 

Figure 3a shows UV-Vis absorption spectra of GO and GO/AgNP composites. As shown in Figure 3a, there is a maximum at 230 nm (indicating the electronic π−π* transitions of aromatic C−C bonds) and a shoulder at about 306 nm (assigned to the n−π* transitions of C−O bonds) in the UV-Vis spectra of GO. After AgNP modification of GO, the UV-Vis spectra of GO-8, GO-20 and GO-33 all have a characteristic peak of AgNP at 398 nm, 406 nm, and 410 nm. The characteristic peak range of AgNP is between 398 nm and 420 nm, and shifts to larger wavelength with increasing size of AgNP, which is consistent with TEM image (Figure 2b–d) and size distribution (Figure 2f–h).The XRD patterns of GO and three types of GO/AgNP composites are shown in Figure 3b. The characteristic peak of GO can be observed at 2θ = 11.02°. The peaks at 2θ = 38.16°, 44.28°, 64.28°, and 77.42° are assigned to the (111), (200), (220), and (311) reflection of Ag.

The morphology and structure of GO and GO/AgNP composite membranes was characterized by SEM. As shown in Figure 4a, there are wrinkles on the surfaces of GO sheets due to oxygen-containing groups. Figure 4c–d shows the top view image of three types of GO/AgNP composite membranes; the different sized of AgNPs can be easily observed in these micrographs. The cross-sectional SEM image of the GO membrane (Figure 4e) shows its layered structure. The cross-sectional SEM image of the three types of GO/AgNP composite membranes are shown in Figure 4f–h. The GO layers are tightly locked between AgNPs, and there are more wrinkles observed in the cross section of all three types of GO/AgNP composite membranes compared to the GO membrane. These additional features provide more nanochannels for water permeation, as discussed below.

The surface roughness of four types of GO membranes were characterized by AFM, as shown in Figure 5. The root-mean-square surface roughness, R_q_, of GO, GO-8, GO-20 and GO-33 were 115 nm, 39 nm, 25.5 nm, and 15.2 nm, respectively. The arithmetic mean height difference, R_a_, of GO, GO-8 GO-20, and GO-33 was 92.5 nm, 30.8 nm, 19.5 nm, and 11.1 nm, respectively. The roughness of the four types of membranes decreases with an increasing size of AgNPs, due to AgNPs filling in the wrinkles and defects of GO sheets. When the size of AgNPs increased, the defects of the GO sheet became narrower, which decreased the surface roughness. Figure 6 is water contact angle of four types of membrane. This shows that the water contact angle enhanced after AgNPs modification, which indicates the composite membrane is more hydrophobic. This is because the decrease in oxygen-containing functional groups which are hydrophilic, due to Ag^+^ combining with these groups in fabrication of GO/AgNPs composite (Scheme 1). With the size of AgNPs increasing, there are more hydrophilic groups combining with Ag^+^. This benefits water molecules flowing, and shows the composite membrane has better performance in terms of permeation ability. 

The influence of thickness of membrane, which was controlled by volume of solution used to prepare the membrane, on water flux was investigated. As shown in Figure 7a, the water flux is negatively correlated to the thickness of four types of membranes. With increase of thickness, the water flux decreases sharply, which suggests lower permeation of membranes. When the volume of solution is 1 mL, the water flux of GO, GO-8, GO-20, and GO-33 is 60.46 L m^−2^ h^−1^ bar^−1^, 99.8 L m^−2^ h^−1^ bar^−1^, 106.1 L m^−2^ h^−1^ bar^−1^ , and 74.8 L m^−2^ h^−1^ bar^−1^, respectively. When the volume of solution increases to 3 mL, the water flux of GO, GO-8, GO-20, and GO-33 is 18.6 L m^−2^ h^−1^ bar^−1^, 21.4 L m^−2^ h^−1^ bar^−1^, 33.9 L m^−2^ h^−1^ bar^−1^ , and 20.8 L m^−2^ h^−1^ bar^−1^, which is reduced by 69.2%, 78.6%, 68%, and 72.2%, respectively, for the four types of membranes. Figure 7a also shows that the GO membrane has the worst performance in water flux. Comparing to the GO membrane, GO-8 has higher water flux at the same thickness. The water flux of GO-20 is higher than that of GO-8. However, when the size of AgNP increases to 33 nm, the water flux declines to a value that is even lower than that of GO-8.

The rejection of RhB of four types of membranes of different thickness are shown in Figure 7b. The rejection is positively correlated to the thickness of the membrane. When the volume of solution is 1 mL, the rejection of GO, GO-8, GO-20, and GO-33 is 99.85%, 77.91%, 85.87%, and 84.16%, respectively. When the membrane thickness is increased by increasing the volume of solution to 3 mL, the rejection of the four types membranes is 99.97%, 91.27%, 97.73%, and 95.61%, which increase by 0.12%, 13.36%, 11.86%, and 11.45%. The influence of the size of AgNP on rejection was also investigated. The rejection of GO-8 is lower than that of GO at same thickness. However, GO-20 has higher rejection compared to GO-8. When the size of AgNPs is 33 nm, the rejection decreases, but is still higher than that of GO-8.

The size sieving effect is purported to be the dominant separation mechanism of nanofiltration membranes [23]. Mi et al. proposed tuning the interlayer spacing by insertion of chemical functional groups, polymers, or nanoparticles, to achieve selective separation [31,32]. Therefore, three types of different sized GO/AgNP composite membranes were successfully fabricated to investigate the size effect of inserted nanoparticles on nanofiltration performance. The water flux is supposed to increase while the rejection decreases upon addition of larger sized AgNPs, which enlarge the interlayer spacing of GO sheets. However, the result was not exactly as expected. Besides interlaying spacing, we conclude that defects in the GO nanosheet should also be regarded as a crucial factor influencing the nanofiltration performance of GO membrane.

Scheme 2 shows a schematic of the proposed separation mechanism of GO/AgNP composite membranes. Water molecules and RhB molecules pass through the defects first, and then RhB molecules are blocked by the nanochannels between GO sheets while water molecules can permeate the membrane. Larger-sized s enlarge the interlayer spacing (d_3_ > d_2_ > d_1_), but decrease the defect size (d’_3_ < d’ _2_ < d’_1_), as demonstrated by AFM roughness measurements. When the size of AgNPs is 8 nm, the spacing between GO sheets is larger than that of GO membrane, however the the defect size is not substantially altered. Therefore, GO-8 has higher water flux and lower rejection than GO. When the size of AgNP increases to 20 nm, the interlayer spacing continues to enlarge, while the defects become narrower. Both water flux and rejection of GO-20 are higher than GO-8. Because water molecules are much smaller than RhB molecules, the narrower defects can selectively reject RhB molecules with larger interlayer spacing increasing water flux. However, the water flux of GO-33 start declines compared to the other membranes, because of the 33 nm AgNPs significantly narrow the defects, thereby reducing the water flux. The rejection of GO-33 also decreases due to the larger interlayer spacing, compared to the GO-20 composite membrane.

Figure 8 shows the BJH adsorption pore distribution spectra of four types of GO membranes. The BET surface area of them was 8.9816 m^2^/g, 18.928 m^2^/g, 26.5457 m^2^/g, and 2.3405, respectively. The AgNPs loaded on GO increased the surface area of the membrane. However, the surface area of GO-33 is smaller than that of GO. The pore width of them was 4.9386 nm, 4.6173 nm, 4.7303 nm, and 6.6469 nm, respectively; pore volume of them was 0.008756 cm^2^/g, 0.01912 m^2^/g, 0.03046 m^2^/g, and 0.004254 m^2^/g. As shown in Figure 8a–c, pore volume is positively correlated to the size of AgNPs. However, when the size of AgNPs increased to 33 nm, pore volume decreased though pore width was larger than that of GO-20. These data provide further support for the decline in filtration ability of GO-33.

The influence of feed concentration on rejection is shown in Figure 9a. The feed concentration of RhB solution is 20 mg/L, 40 mg/L, and 60 mg/L, respectively. The rejection is negatively correlated to the feed concentration of RhB solution. With the feed concentration increases, the rejection rate declines sharply. When the feed concentration is 20 mg/L, the rejection rate of GO-8, GO-20, and GO-33 is 85.99%, 96.00%, and 91.53%, respectively. When it increases to 60 mg/L, the rejection of three types of composite membranes drop to 49.40%, 57.36%, and 50.16%, which decreases by 36.59%, 38.64%, and 41.37%. Higher feed concentration increases the permeation speed of RhB, resulting in stronger solute diffusion effect, which causes lower rejection during nanofiltration progress. The flux of common solvents through GO-20 was investigated in Figure 9b. The viscosity of methanol, ethanol and ethylene glycol are 0.60 mPa·s, 1.17 mPa·s, and 19.9 mPa·s, respectively. The flux of methanol, ethanol and ethylene glycol is 48.33 L m^−2^ h^−1^ bar^−1^, 23.02 L m^−2^ h^−1^ bar^−1^, and 7.01 L m^−2^ h^−1^ bar^−1^. The flux of solvents decreases with increasing viscosity, which is consistent with results of a previous study [22].

The rejection of three types of GO composite membranes for both RhB and MB are shown in Table 1. GO-20 has the highest rejection rate of RhB and MB, which as shown above. Comparing to RhB, the rejection of MB using GO-8, GO-20, and GO-33 increases by 16.67%, 12.63%, and 12.61%, respectively. Besides the size exclusion effect, electrostatic repulsion influences nanofiltration performance as well [33]. The size of MB molecules is larger than the size of RhB molecules. RhB is positively charged dye, while MB is negatively charged dye. Therefore, GO composite membranes have a higher rejection of MB than RhB, because of the negatively oxygen-containing groups on GO. Table 2 summarizes other recent reports of graphene oxide composite membranes and their filtration performance. All three types of GO/AgNPs composite membranes reported here have high water permeation ability and high rejection, comparable to these other recent reports.

## 4. Conclusions

In summary, three types of GO/AgNP composite membranes with different sized AgNPs were fabricated. The morphology and XRD results show that AgNPs were successfully loaded into GO membranes, and the size of AgNPs was 8 nm, 20 nm, and 33 nm. The water flux of GO/AgNP composite membranes and GO membranes is negatively correlated to the thickness of membrane, while the rejection of four types membranes is positively correlated to membrane thickness. When the size of AgNPs is 20 nm, the GO/AgNPs composite membrane has the highest water flux and rejection among three types of composite membrane. This performance is related to both interlayer spacing, which increases with increasing AgNP size, and to defect size, which decreases with increasing AgNP size. The rejection is negatively correlated to feed concentration of RhB solution due to the solute diffusion effect. The rejection of MB for GO/AgNP composite membranes and the GO membrane is higher than that of RhB, due to physical size sieving and electrostatic repulsion mechanisms. This work indicates that both interlayer spacing and defect size influence the nanofiltration performance, suggesting two competing mechanisms whereby nanosized additives affect membrane performance.

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
