# Peer review of "Graphene Oxide Nanofiltration Membranes Containing Silver Nanoparticles: Tuning Separation Efficiency via Nanoparticle Size"

_nanomaterials, 2020, doi:10.3390/nano10030454_

Round 1

Reviewer 1 Report

The authors investigated the effect of decoration of the membranes with Ag nanoparticles (AgNP) possessing different sizes. To understand the role of the Ag NP authors performed different chemical morphological structural analyses. The description of these analyses can be improved better discussing the results. Outcomes from  permeation/rejection experiments were interpreted on the basis of coupled effects arising from separation of GO layers induced by the NPs, defect extension and chemical interactions.

The abstract should describe the aim of the manuscript. Here authors did not clarify why they infiltrated the membranes with nanoparticles why they selected Ag nanoparticles among the plethora of possible other nanostructured materials. Similar comment holds also for the introduction. Apparently it seems that Ag NP were selected just to modify the spacing between GO layers. This could be done also using less expensive materials. Authors refer to the work of Chen et al. ref [27] who found that Ag possesses antibacterial properties. This property is not new and is common to other metallic NP. However, the focus of the manuscript, as it appears from the first part of the introduction, regards the degradation of organic molecules such as dyes and not the antibacterial activity. Then still the question why Ag was selected as spacer for GO membranes remains open. Please better clarify this point. Section 2.2 materials and methods: the authors synthesized GO/AgNP composite with AgNP of 8nm using 25mg AgNO3. Authors should better describe what is presented in scheme 1. They added 50 mg or 100 mg of AgNO3 to the suspension. I supposes the higher concentration of AgNO3 leads to an increase of the AgNP.  In addition, following scheme 1, the temperature leads to a reduction of the GO. Please better describe the synthesis procedure. Fabrication of the membranes: the authors utilized the vacuum filtration method to produce the membranes. They provided poor information about the fabrication process. It is not clear if the cellulose acetate used as support will be or not part of the final product. Is also not clear if the authors just dried the GO/AgNP in air or in inert atmosphere, if they applied or not heating. In addition figure S2 shows two membranes obtained using pure GO or GO/AgNP but they did not specified which kind of AgNP: 8 nm, 20 nm or 33 nm? It would be interesting to show all the membranes obtained with the different AgNPs. Section 2.4 Characterization. The authors used FTIR to characterize the chemistry of the membranes. In the article abstract authors declare they characterized their membranes using XPS. However, in section 2.4 XPS is not mentioned among the techniques used to obtain the chemical composition of the membranes... Section 3 Results and discussion: authors state: “ The FTIR spectra of GO/AgNPs composite shows that the peaks at 3430, 1740 and 1050 cm−1 decrease after AgNP modification, due to interaction of Ag+ ions and carboxyl groups on the edges of GO sheets”… FTIR spectra of GO/AgNP show peaks in different energy position which in principle could be associated to different bonds. Authors do not explain the reason of this energy shift, if it is induced by Ag and why. Raman spectroscopy: ref. 31 is a wrong reference. In Ref 31 Sun et al. show spectra only in the range 800 – 2000 cm-1. These authors relate the presence of surface enhanced Raman scatterning to the much higher intensities of the D and G bands in GO/AgNP sample with respect to those of the pure GO sample. Please provide a correct reference and better discuss the band at 2840cm-1. UV-vis characterization. The authors could estimate the AgNP size using the Mie theory and correlate with TEM analyses. Water contact angle. The authors found that increasing the AgNP content increased the hydrophobicity of the membrane. Have they any idea for this effect? Looking at SEM inages it appears that almost all the surface is covered by Ag NP. It is known metallic surfaces are characterized by high energy which is in contrast with the results obtained. Any effect induced by the surface morphology? English should be revised and some grammar errors need correction.

Author Response

Point 1: The abstract should describe the aim of the manuscript. Here authors did not clarify why they infiltrated the membranes with nanoparticles why they selected Ag nanoparticles among the plethora of possible other nanostructured materials. Similar comment holds also for the introduction. Apparently it seems that AgNP were selected just to modify the spacing between GO layers. This could be done also using less expensive materials. Authors refer to the work of Chen et al. ref [27] who found that Ag possesses antibacterial properties. This property is not new and is common to other metallic NP. However, the focus of the manuscript, as it appears from the first part of the introduction, regards the degradation of organic molecules such as dyes and not the antibacterial activity. Then still the question why Ag was selected as spacer for GO membranes remains open. Please better clarify this point.

Response 1: It’s a good question. Because we have investigated how the amount of AgNPs loaded on GO influences membrane nanofiltration membrane, and the fabrication of different-sized AgNPs is possible. Therefore, we chose AgNPs as spacer. We explained this point:

“We have investigated the influence of the amount of AgNPs on membrane nanofiltration performance[32]. The fabrication method of different-sized AgNPs is mature[27].Therefore, in this work, we chose AgNPs as spacer, and successfully fabricated three types of GO/AgNP composite membranes.”

Point 2: Section 2.2 materials and methods: the authors synthesized GO/AgNP composite with AgNP of 8nm using 25mg AgNO3. Authors should better describe what is presented in scheme 1. They added 50 mg or 100 mg of AgNO3 to the suspension. I supposes the higher concentration of AgNO3 leads to an increase of the AgNP. In addition, following scheme 1, the temperature leads to a reduction of the GO. Please better describe the synthesis procedure.

Response 2: The concentration of AgNO3 can influence the size of AgNP, but the synthesis procedure is not clear enough. We changed another scheme to make procedure easier to understand:

Point 3: Fabrication of the membranes: the authors utilized the vacuum filtration method to produce the membranes. They provided poor information about the fabrication process. It is not clear if the cellulose acetate used as support will be or not part of the final product. Is also not clear if the authors just dried the GO/AgNP in air or in inert atmosphere, if they applied or not heating.

Response 3: The CA membrane is a part of the final product, and the composite membrane was dried in air. We add explanation in this part:

  “A 0.22μm cellulose acetate was used as support membrane which will be part of final product. The composite solution was filtrated through CA membrane under a transmembrane pressure of 1 bar (0.1 MPa), and then dried the composite membrane in air.”

Point4 : In addition figure S2 shows two membranes obtained using pure GO or GO/AgNP but they did not specified which kind of AgNP: 8 nm, 20 nm or 33 nm? It would be interesting to show all the membranes obtained with the different AgNPs.

Response 4: Thanks for your suggestion. As we take pictures with a low resolution digital camera, all types of GO/AgNPs composite membrane look very similar to each other. The two figure show the different of composite membrane and GO membrane. Due to the impact of our country's epidemic situation, back to school is currently not allowed and may last for a long time. Therefore ,it ‘s difficult for us to obtain the other pictures.

Point 5: Section 2.4 Characterization. The authors used FTIR to characterize the chemistry of the membranes. In the article abstract authors declare they characterized their membranes using XPS. However, in section 2.4 XPS is not mentioned among the techniques used to obtain the chemical composition of the membranes.

Response 5: XPS mentioned in abstract was a written error. The characterization we used was XRD. We have fixed it .

Point 6: Section 3 Results and discussion: authors state: “ The FTIR spectra of GO/AgNPs composite shows that the peaks at 3430, 1740 and 1050 cm−1 decrease after AgNP modification, due to interaction of Ag+ ions and carboxyl groups on the edges of GO sheets”… FTIR spectra of GO/AgNP show peaks in different energy . position which in principle could be associated to different bonds. Authors do not explain the reason of this energy shift, if it is induced by Ag and why.

Response 6: It’ s a good suggestion. There are not enough explanation about FTIR spectra, so we added it in this part:

 “ Some prominent peaks shifted after AgNPs modification. Because the strongly interaction between Ag+ ions and carboxyl groups on the edges of GO sheets, the intensity of −OH decrease sharply, and both intensity and peak position of C=O group changed, as shown in the FTIR spectra of GO/AgNPs composite. The intensity and peak position of C−O−C and C−O groups also changed, which indicated the vast of oxygen-containing functional groups contribute to the proper decoration of AgNPs on GO sheets. Compare to GO, GO/AgNPs composite have obvious change in peak position and intensity, which shows composite does not produce new chemical bonds, but due to the electrostatic adsorption. ”

Point 7:Raman spectroscopy: ref. 31 is a wrong reference. In Ref 31 Sun et al. show spectra only in the range 800 – 2000 cm-1. These authors relate the presence of surface enhanced Raman scatterning to the much higher intensities of the D and G bands in GO/AgNP sample with respect to those of the pure GO sample. Please provide a correct reference and better discuss the band at 2840cm-1.

Response 7: The reference is right, but the explanation is not clear enough, and there are some mistake in figure. 1b. We improved this part and changed figure. 1b:

“ Fig. 1b shows Raman spectra of GO and GO/AgNP composites. Comparing to GO, the D and G band obviously enhanced after AgNP modification, because of surface enhanced Raman scattering from the forceful local electromagnetic fields of AgNPs along with the plasmon resonance. The intensity ratio of GO/AgNPs composite (ID/IG=1.02) is also higher than that of GO (ID/IG=0.92), indicating a decrease in crystallite size of GO−AgNPs composite. [31]”

Point 8: UV-vis characterization. The authors could estimate the AgNP size using the Mie theory and correlate with TEM analyses. Water contact angle. The authors found that increasing the AgNP content increased the hydrophobicity of the membrane. Have they any idea for this effect?

Response 8: It’ s necessary to connect characterize and performance, and we discussed about it:

“ Fig. 6 is water contact angle of four types of membrane. It shows that water contact angle enhanced after AgNPs modification, which indicates the composite membrane is more hydrophobic. This benefits to water molecule flowing, and shows the composite membrane has better performance in permeation ability.”

Point 9: Looking at SEM inages it appears that almost all the surface is covered by Ag NP. It is known metallic surfaces are characterized by high energy which is in contrast with the results obtained. Any effect induced by the surface morphology? English should be revised and some grammar errors need correction.

Response 9: Yes, the reviewers observed it very carefully. In the SEM image (fig. 4 b-d), there are lots of silver nanoparticles on the surface of GO. Due to the large surface area and the proportion of surface atoms, it has a high energy state and will exhibit surface effects, size effects, and quantum effects, which will play a role in catalysis, photoelectricity, electromagnetic, etc. In this study, the role of silver particles is to act as an intercalation agent between GO sheets to adjust the interlayer distance, and also to prevent the agglomeration of GO. We have not investigated and studied other effects. Thank you very much for your suggestion, we will pay attention to this impact in the future.

Reviewer 2 Report

The article can be published after the following changes:

The discussion of the results is too weak and they are not compared with others results presented in papers using similar methods and graphene oxide. The discussion part must be re-written with in systematic scientific manner with enough experimental data and references supporting the ideas. summary formulas of chemical compounds should be written correctly. Eg.
ethanol (C2H6O), methanol (CH4O) are C2H5OH and CH3OH Authors should explain all abbreviations used in the text. Formulas should be numbered and units should be given. Use SI unit throughout the manuscript. Authors should clearly explain transport mechanism. Some grammatical and typing errors should be corrected in the manuscript. Also the whole text calls for a minor edition by the native English speaker.

Author Response

Point 1: The discussion of the results is too weak and they are not compared with others results presented in papers using similar methods and graphene oxide. The discussion part must be re-written with in systematic scientific manner with enough experimental data and references supporting the ideas.

Response 1: More explanation was added to the discussion part:

FTIR:”The peak at 3430 cm−1 corresponds to −OH stretching band. The peaks at 1740, 1610, 1230 and 1050 cm−1 are assigned to C=O stretching vibration, C=C vibration, C−O−C stretching and C−O stretching, which indicate successful synthesis of GO. Some prominent peaks shifted after AgNPs modification. Because the strongly interaction between Ag+ ions and carboxyl groups on the edges of GO sheets, the intensity of −OH decrease sharply, and both intensity and peak position of C=O group changed, as shown in the FTIR spectra of GO/AgNPs composite. The intensity and peak position of C−O−C and C−O groups also changed, which indicated the vast of oxygen-containing functional groups contribute to the proper decoration of AgNPs on GO sheets. Compare to GO, GO/AgNPs composite have obvious change in peak position and intensity, which shows composite does not produce new chemical bonds, but due to the electrostatic adsorption. ”

Raman:” Fig. 1b shows Raman spectra of GO and GO/AgNP composites. Comparing to GO, the D and G band obviously enhanced after AgNP modification, because of surface enhanced Raman scattering from the forceful local electromagnetic fields of AgNPs along with the plasmon resonance.  The intensity ratio of GO/AgNPs composite (ID/IG=1.02) is also higher than that of GO (ID/IG=0.92), indicating a decrease in crystallite size of GO−AgNPs composite..[31]”

Water contact angle:” Fig. 6 is water contact angle of four types of membrane. It shows that water contact angle enhanced after AgNPs modification, which indicates the composite membrane is more hydrophobic. This benefits to water molecule flowing, and shows the composite membrane has better performance in permeation ability. ”

Point 2: summary formulas of chemical compounds should be written correctly. Eg.ethanol (C2H6O), methanol (CH4O) are C2H5OH and CH3OH.Authors should explain all abbreviations used in the text.

Response 2: The written error has been corrected, and all abbreviations was explained.

Point 3: Formulas should be numbered and units should be given. Use SI unit throughout the manuscript.

Response 3: Formulas were numbered and units were given.

                                                                (1)

Where V (L) is the volume of water, A (m2) is the effective filtration area of the membrane, t (h) is the filtration time, and P (bar) is the transmembrane pressure.

Rhodamine B (RhB) and methylene blue (MB) were used to characterize the rejection performance of composite membrane. The rejection can be calculated using the following equation:

                                                         (2)

Where A2 is the absorption of the filtrate, and A1 is the absorption of the original solution.

Point 4: Authors should clearly explain transport mechanism.

Response 4: The transport mechanism was explained:

“Scheme 2 shows a schematic of the proposed separation mechanism of GO/AgNP composite membranes. Water molecules and RhB molecules pass through the defects first, and then RhB molecules are blocked by the nanochannels between GO sheets while water molecules can permeate the membrane. Larger-sized s enlarge the interlayer spacing (d3>d2>d1), but decrease the defect size (d’3<d’2<d’1), as demonstrated by AFM roughness measurements. When the size of AgNPs is 8nm, the spacing between GO sheets is larger than that of GO membrane, however the the defect size is not substantially altered. Therefore, GO-8 has higher water flux and lower rejection than GO. When the size of AgNP increases to 20nm, the interlayer spacing continues to enlarge, while the defects become narrower. Both water flux and rejection of GO-20 are higher than GO-8. Because water molecules are much smaller than RhB molecules, the narrower defects can selectively reject RhB molecules with larger interlayer spacing increasing water flux. However, the water flux of GO-33 start declines compared to the other membranes, because of the 33nm AgNPs significantly narrow the defects, thereby reducing the water flux. The rejection of GO-33 also decreases due to the larger interlayer spacing, compared to the GO-20 composite membrane.”

Scheme 2 Schematic diagram indicating the separation mechanism of (a) GO-8 (b) GO-20 (c) GO-33 membranes.

Point 5: Some grammatical and typing errors should be corrected in the manuscript. Also the whole text calls for a minor edition by the native English speaker.

Response 5: We have edited the grammar and spell carefully.

Reviewer 3 Report

This study shows graphene oxide(GO)/ silver nanoparticles composite membranes used for separation of Rhodamine B and methylene blue according to the size effect of Ag nanoparticles. Some interesting data’s are available.

X-ray photoelectron spectroscopy mentioned in abstract I could not found any XPS data. Introduction part needs more explanation about the separation applications. Whether, the composite membrane is loose nanofiltration or tight nanofiltration. What is pore size of prepared membranes ? How the author controlled the agglomeration of nanoparticles ? In TEM images (figure 2), the nanoparticles clusters was found. Figure 2 b (TEM image) is not clear, provide the clear image or high resolution image. Figure 8 has lack of information, needs more explanation. Add the reference from “Nanomaterials” Journal

Author Response

Point 1: X-ray photoelectron spectroscopy mentioned in abstract I could not found any XPS data.

Response 1: XPS mentioned in abstract was a written error. The characterization we used was XRD. We have fixed it .

Point 2: Introduction part needs more explanation about the separation applications.

Response 2: We added more explanation about separation applications:

“Joshi et al. reported that ions smaller in size than nanochannels can easily pass through GO membranes, and that size exclusion is the dominant sieving mechanism. [22] Tuning the interlayer spacing of GO sheets is a promising strategy for improving water flux of GO membranes. [23] Interlayer spacing can be altered via incorporation of silver nanoparticles (AgNPs), to make GO/AgNP composite membranes. [24,25] Sun et al. [26] investigated the influence of the thickness of GO/AgNPs membrane to permeability. The result showed that the water flux increased with reduction of the membrane thickness. They also investigated the water contact angle of GO and GO/AgNPs membrane. The result showed that the water contact angle of GO/AgNPs membrane was larger than that of GO membrane, which indicated GO/AgNPs membrane is more hydrophobic. Liu et al. investigated influence of membrane thickness to rejection to RhB. The result showed membrane thickness positively correlated to rejection. [32] These studies indicate membrane thickness is critical to nanofiltration performance. Chen et al. [27] found a negative correlation between the size of AgNPs in GO@PEG composite membranes and antibacterial activities. However, there are few studies about how the size of AgNPs on GO membranes effects the permeability and rejection of organic dye.”

Point 3: Whether, the composite membrane is loose nanofiltration or tight nanofiltration. What is pore size of prepared membranes ?

Response 3: The pore size of the membrane was measured by BET. Pore width of them was 4.9386nm, 4.6173nm, 4.7303nm and 6.6469nm, respectively; pore volume of them was 0.008756cm2/g, 0.01912m2/g, 0.03046m2/g, 0.004254m2/g.

Point 4: How the author controlled the agglomeration of nanoparticles ? In TEM images (figure 2), the nanoparticles clusters was found.

Response 4: PVP solution was used to prevent the agglomeration of nanoparticles, which was mentioned in experiment part.

Point 5: Figure 2 b (TEM image) is not clear, provide the clear image or high resolution image.

Response 5: We changed fig.2b. The new one is clearer.

Point 6: Figure 8 has lack of information, needs more explanation.

Response 6: We agree that there are not enough explanation about this figure, so we added more detail in this part:

   “ The BET surface area of them was 8.9816m2/g, 18.928m2/g, 26.5457m2/g and 2.3405m2/g, respectively. The AgNPs loaded on GO increased the surface area of the membrane. But the surface area of GO-33 is smaller than that of GO. Pore width of  them was 4.9386nm, 4.6173nm, 4.7303nm and 6.6469nm, respectively; pore volume of them was 0.008756cm2/g, 0.01912m2/g, 0.03046m2/g, 0.004254m2/g.”

Point 7: Add the reference from “Nanomaterials” Journal

Response 7: We have added reference from Nanomaterials:

[37] Plachá, D.; Jampilek, J. Graphenic materials for biomedical applications. Nanomaterials. 2019, 9, https://doi.org/10.3390/nano9121758.

[38] Wang, Y.; Guo, L.; Qi, P. F.; Liu, X. M.; Wei, G. Synthesis of three-dimensional graphene-based hybrid materials for water purification: a review. Nanomaterials. 2019, 9, https://doi.org/10.3390/nano9081123

Round 2

Reviewer 1 Report

1) The Raman spectrum needs a better description. The Id/Ig ratio increases in the case of the decorated Graphene membrane. The authors justify the more intense D band as deriving from an increased crystallite size. It is not clear which are the crystallites. Do they refer to graphene? In this case why decoration should lead to a reduction of the crystalline domains in graphene flakes? On the other hand, D band refers to the system disorder…

2) The authors did not address the point regarding the UV-vis characterization and in particular the possibility to estimate the AgNP size using the Mie theory and correlate with TEM analyses.

3) Authors also did not addressed the question regarding the contact angle and why it increases increasing the Ag NP dimensions.

Author Response

Point 1: The Raman spectrum needs a better description. The Id/Ig ratio increases in the case of the decorated Graphene membrane. The authors justify the more intense D band as deriving from an increased crystallite size. It is not clear which are the crystallites. Do they refer to graphene? In this case why decoration should lead to a reduction of the crystalline domains in graphene flakes? On the other hand, D band refers to the system disorder…

Response 1: The increased crystallite size refers to the average crystallite size of GO-AgNPs composite. We added the reason to Raman description:

“The intensity ratio of GO/AgNPs composite (ID/IG=1.02) is also higher than that of GO (ID/IG=0.92), indicating a decrease in the average crystallite size of GO−AgNPs composite, due to more defects in graphene composite material after AgNPs modification. [31] The degree of disorder decreased because of the average crystallite size of GO-AgNPs composite decreasing. ID/IG ratio is inversely proportional to the average crystallite size and the degree of disorder.”

Point 2: The authors did not address the point regarding the UV-vis characterization and in particular the possibility to estimate the AgNP size using the Mie theory and correlate with TEM analyses.

Response 2: We combined TEM image and size distribution:

“The TEM image and size distribution of three types of GO/AgNP composites are shown in Figs. 2b-d and f-h. The sizes of AgNPs loaded on the three types of GO/AgNP composites are 8nm, 20nm and 33nm, respectively, as shown in Fig2. f-h, which is consistent with TEM image (Fig. 2b-d ).”

Besides, we also combined UV-vis with TEM analyses in this part:

“After AgNP modification of GO, the UV-Vis spectra of GO-8, GO-20 and GO-33 all have a characteristic peak of AgNP at 398nm, 406nm and 410nm. The characteristic peak range of AgNP is between 398nm and 420nm, and shifts to larger wavelength with increasing size of AgNP, which is consistent with TEM image (fig. 2b-d) and size distribution (fig.2f-h).”

Point 3: Authors also did not addressed the question regarding the contact angle and why it increases increasing the AgNP dimensions.

Response 3: We explained the relationship between the contact angle and the size of AgNPs:

“It shows that water contact angle enhanced after AgNPs modification, which indicates the composite membrane is more hydrophobic. This is because the decrease in oxygen-containing functional groups which are hydrophilic, due to Ag+ combining with these groups in fabrication of GO/AgNPs composite.(Scheme 1) With the size of AgNPs increasing, there are more hydrophilic groups combining with Ag+. This benefits to water molecule flowing, and shows the composite membrane has better performance in permeation ability. ”

Reviewer 2 Report

The article can be published in this Journal, after the introduced corrections.

Author Response

Point: The article can be published in this Journal, after the introduced corrections.

Response :Thank you for your comments and suggestions. We have modified the article carefully.